# EFFICIENT AND STEALTHY BACKDOOR ATTACK TRIGGERS ARE CLOSE AT HAND

## ABSTRACT

A backdoor attack aims to inject a backdoor into a deep model so that the model performs normally on benign samples while maliciously predicting the input as the attacker-defined target class when the backdoor is activated by a predefined trigger pattern. Most existing backdoor attacks use a pattern that rarely occurs in benign data as the trigger pattern. In this way, the impact of the attack on the label prediction of benign data can be mitigated. However, this practice also results in the attack being defended against with little performance degradation on benign data by preventing the trigger pattern from being activated. In this work, we present a new attack strategy to solve this dilemma. Unlike the conventional strategy, our strategy extracts the trigger pattern from benign training data, which frequently occurs in samples of the target class but rarely occurs in samples of the other classes. Compared with the prevailing strategy, our proposed strategy has two advantages. First, it can improve the efficiency of the attack because learning on benign samples of the target class can facilitate the fitting of the trigger pattern. Second, it increases the difficulty or cost of identifying the trigger pattern and preventing its activation, since many benign samples of the target class contain the trigger pattern. We empirically evaluate our strategy on four benchmark datasets. The experimental studies show that attacks performed with our strategy can achieve much better performance when poisoning only 0.1% or more of the training data, and can achieve better performance against several benchmark defense algorithms.

## 1 INTRODUCTION

Backdoor attack, also known as Trojan horse attacks, has become an increasing security threat in recent years, attracting many research interests (Chen et al., 2017; Doan et al., 2021). The attack aims to inject a backdoor into a deep model so that the model behaves normally on benign samples, while its predictions are maliciously and consistently changed to a predefined target class (or classes) when the backdoors are activated. Currently, poisoning training samples is the most straightforward and widely adopted method for performing backdoor attacks. Depending on whether the labels of the poisoned samples are changed or not, existing backdoor attacks can be roughly divided into poison-label backdoor attacks (Gu et al., 2017; Barni et al., 2019; Nguyen & Tran, 2020; Liu et al., 2020; Qi et al., 2021) and clean-label backdoor attacks (Turner et al., 2019; Saha et al., 2020).

In this work, we follow the practice of poison-label backdoor attacks, which are much more efficient than clean-label ones. To perform a poison-label backdoor attack, it first selects a small number (a smaller number means a smaller impact on the performance of the benign data and a lower probability of being inspected by the developer) of benign samples from the training set, inserts a predefined trigger pattern into the inputs, and changes their labels to the target class (or classes). It then reinserts the poisoned samples into the training set and provides the resulting data to the victims to train the model with. In attack mode, it manipulates the victim model to produce the intended target class by injecting the predefined trigger pattern into the inputs.

Existing backdoor attacks typically use a pattern that rarely occurs in benign data as the trigger pattern. For example, Gu et al. (2017) used stickers and checkerboards as trigger patterns for image data, Sun (2020) used special characters, words, and phrases as trigger patterns for text data. This strategy can prevent the attack from being falsely activated on benign data and mitigate the impact of

the attack on the label predictions of benign samples. However, it also results in low-effort detection and mitigation of the attack by preventing activation of the trigger pattern. As far as we know, most existing backdoor attack defense algorithms build on this property and achieve great success (Chen et al., 2019; Wang et al., 2019; Zhao et al., 2020; Xu et al., 2020; Yang et al., 2021; Huang et al., 2022a; Mu et al., 2022).

In this work, we introduce a new attack strategy to solve this dilemma. Instead of using a rare pattern as the trigger pattern, we extract the trigger pattern from benign data that frequently occurs in samples of the target class but rarely occurs in samples of non-target classes. Accordingly, we introduce a framework to extract the trigger pattern and insert it into benign inputs to perform data poisoning. The proposed strategy has two advantages over the prevailing one. **First**, it is more efficient since the trigger pattern comes from the target class. Training on benign samples of the target class will help the fitting of the trigger pattern. **Second**, it becomes harder to detect and defend against the attack because many benign samples of the target class contain the trigger pattern. It is difficult to distinguish the poisoned samples from benign samples of the target class using the trigger pattern, and disabling the activation of the trigger pattern will degrade performance of these benign target class samples.

To evaluate the benefits of our proposed strategy, we conducted experiments on four widely studied datasets. The empirical results show that attacks performed with our strategy can achieve better attack performance with less poisoned data especially when the poisoned data size is small. Moreover, they can escape the defenses of several benchmark defense algorithms, while attacks performed with the conventional strategies often fail to do so.

The main contributions of this work can be summarized in three points. **1)** A new attack strategy for designing the trigger pattern is proposed. **2)** An effective technique for extracting the trigger pattern and injecting it into benign data is proposed. **3)** Experiments are conducted on four widely studied datasets, and the results show that our strategy can improve both attack efficiency and stealthiness.

## 2 BACKGROUND & RELATED WORK

### 2.1 BACKDOOR ATTACK

**Threat Model.** Backdoor attacks aim to insert backdoors into deep models so that victim models will work genuinely on benign inputs but misbehave when a specific trigger pattern occurs. This is an emerging area of research that raises security concerns about training with third-party resources. A backdoor attack can be performed at multiple stages of the victim model (Li et al., 2022a; Jia et al., 2022; Chen et al., 2022). *In this work, we follow the widely used training data poisoning setting, where attackers only manipulate the training data, but not the model, training schedule, or inference pipeline*. In this case, the attackers spread the attack by sending the poisoned data to the victims or publishing it on the Internet and waiting for the victims.

Formally, let $\boldsymbol{x} \in \mathcal{X}$ denote the input, $y \in \mathcal{Y}$ denote the output, $\mathcal{D} = \{(\boldsymbol{x}_1, y_1), \cdots, (\boldsymbol{x}_n, y_n)\}$ denote the benign training dataset. To perform a backdoor attack, the attacker first selects a small number of benign samples from $\mathcal{D}$ and then injects the predefined trigger pattern into each selected input and sets its label as follows:
$$\tilde{\boldsymbol{x}} = \mathcal{B}(\boldsymbol{x}); \tilde{y} = T(y),$$
where $\mathcal{B}$ denotes the backdoor injection function and $T$ the relabel function. In the usual **all-to-one** poison-label attack setting (we apply the setting in this work), $T(y) \equiv c_t$, where $c_t$ denotes the label of the attacker-specific target class. Then, the attacker mixes the poisoned samples, $(\tilde{\boldsymbol{x}}, \tilde{y})$, with the remaining benign training samples and provides the resulting dataset, $\tilde{\mathcal{D}}$, to the victims. Without knowing the training data being poisoned, the victims train a model, $f_{vim}$, on $\tilde{\mathcal{D}}$, and provide a classification service using $f_{vim}$. In attack mode, the attacker injects the trigger pattern into any input $\boldsymbol{x}$ that they want the victim model to behave as follows:

$$f_{vim}(\boldsymbol{x}) = y; f_{vim}(\mathcal{B}(\boldsymbol{x})) = T(y).$$

**Previous Poison-Label Backdoor Attacks.** Existing poison-label backdoor attacks typically use a pattern that rarely occurs in benign data as the trigger pattern so as not to degrade the model's performance on benign data (Liu et al., 2017; Chen et al., 2017; Barni et al., 2019; Liu et al., 2020).

As a pioneering work, **BadNet** (Gu et al., 2017) has used stickers and checkerboard as trigger patterns for image data. It replaces a fixed area of the input image with the trigger pattern to poison the input image, i.e., $\mathcal{B}(\boldsymbol{x}) = \boldsymbol{m} \odot \boldsymbol{x} + (1 - \boldsymbol{m}) \odot \boldsymbol{\eta}$, where $\boldsymbol{m}$ is a binary mask matrix and $\boldsymbol{\eta}$ is an image with checkerboards in the corners. After that, many variants of backdoor attacks were introduced. Blend (Chen et al., 2017) used a normal image as the trigger pattern and mixed the trigger pattern with the benign input to inject the backdoor. **WaNet** (Nguyen & Tran, 2021) proposed using a small and smooth warping field as the trigger pattern to poison images. (Sun, 2020) proposed injecting special characters, words, and phrases into text data as trigger patterns. And (Qi et al., 2021) poisoned the text inputs to follow a certain syntactic rule, which rarely occurs in benign data. Recently, **ISSBA** (Li et al., 2021c) proposed an instance-specific attack strategy that generates a specific trigger pattern for each of the poisoned inputs. By itself, ISSBA's trigger pattern can be viewed as a combination of the benign input and a predefined encoder-generated trigger string that never occurs in benign data.

**Backdoor Attack Evaluation.** Backdoor attacks are usually evaluated based on two dimensions, i.e., attack *efficiency* and *stealthiness*. In classification tasks, the accuracy on the poisoned test dataset (all samples in the test dataset are poisoned), called the attack success rate (**ASR**), is often used to evaluate attack efficiency. And a simple measure to evaluate the stealthiness of an attack is the classification accuracy on benign test dataset, called benign accuracy (**BA**). A stealthy attack should achieve a BA that is close to that of an intact model, otherwise the developer will notice the attack or not deploy the victim model. You may also evaluate the stealthiness of an attack based on the poison rate. A stealthy attack should have a small poison rate, so the poisoned sample is unlikely to be inspected by the developer. In addition, the stealthiness of an attack is often evaluated by testing the attack performance against defense algorithms. In this case, the measurement differs by defense algorithm.

## 2.2 BACKDOOR ATTACK DEFENSE

In recent years, several categories of defense methods have been developed to defend against backdoor attacks, including explicit backdoor mining-based methods (Chen et al., 2019; Wang et al., 2019; Liu et al., 2019), implicit poisoned input detection-based methods (Gao et al., 2019; Xu et al., 2020; Yang et al., 2021; Huang et al., 2022b), and model mitigation-based methods (Liu et al., 2018; Zhao et al., 2020; Huang et al., 2022a; Mu et al., 2022; Li et al., 2021a). A unique feature of many of these methods is that they perform defense by preventing the activation of the trigger pattern, e.g., by removing or deactivating the corresponding neurons of the trigger pattern from the neural model or selectively removing samples containing the trigger pattern from the training data.

As a concrete example, Liu et al. (2018) observed that some neurons of the victim model would only be activated when the trigger pattern appeared. Therefore, they proposed a so-called Fine-Pruning method to disable the backdoor behavior by eliminating neurons that were dormant on benign inputs. Xu et al. (2020) applied a similar idea to perform the defense but selected the neurons to prune by the $\ell_\infty$ norm values of neurons. Motivated by similar observations of (Liu et al., 2018), Li et al. (2021a) proposed a so-called **NAD** method to defense the attack. The method introduces a teacher model by finetuning the victim model with a few benign samples and then aligns the neurons of the victim model with those of the teacher model through a neural attention distillation process. This way, it hopes to prevent the activation of the trigger pattern. Gao et al. (2019) observed that the label prediction on the poisoned inputs will almost not change over perturbations. Accordingly, they proposed a perturbation based strategy, called **STRIP**, to defense the attack, which perturbed the input samples with noise and removed samples with small prediction variation given by the victim model. Wang et al. (2019) assumed that the trigger pattern was patch-based and proposed a defense algorithm called **Neural-Cleanse**. For each class, it optimized a patch pattern to make the model classify the input injected by the pattern as the class. If a class yielded a significantly smaller pattern, it considered the patch the trigger pattern and claimed the model was under attack. However, with the development of backdoor attacks, the assumption of this method does not hold for most SOTA backdoor attacks. Li et al. (2021b) found that the attack performance can degrade greatly if the appearance or position of the trigger is changed slightly. Based on this observation, they proposed to use spatial transformations (e.g., shrinking (**ShrinkPad**), flipping) for attack defense. At the time of inference, transformation-based preprocessing was first applied to the test input, and then the processed input was fed into the model to perform label prediction. Recently, Huang et al.

(2022b) proposed a so-called **DBD** algorithm to perform attack defense. The algorithm is based on the motivation that the poisoned samples are hard samples for the learning model. It proposed to first train a feature map using self-supervised learning, then train the victim model on the resulting feature space, and finally re-train the model using the top 50% credible training samples selected by the second-step model. Here, the first step is to enhance the input difference between poisoned samples and benign samples of the target class, and the second step is to find out the poisoned samples, leveraging the fact that deep models tend to first fit easy patterns and then hard ones (Arpit et al., 2017; Zhang et al., 2017).

## 3   PROPOSED APPROACH

### 3.1   OVERVIEW

We follow the scheme of BadNet (Gu et al., 2017) and poison a subset of the training dataset $\mathcal{D}$ to perform an attack under the all-to-one setting. Each selected sample $(\boldsymbol{x}, y)$ is replaced with a poisoned sample $(\mathcal{B}(\boldsymbol{x}), c_t)$, where $\mathcal{B}(\cdot)$ is the backdoor injection function and $c_t$ is the attacker-defined target class label.

The novelty of our strategy lies in the implementation of $\mathcal{B}(\cdot)$, which is roughly defined as follows:

$$\mathcal{B}(\boldsymbol{x}) = f_{dec}\left((1 - \boldsymbol{m}) \odot \boldsymbol{h} + \boldsymbol{m} \odot \boldsymbol{h}_t\right), \text{ with } \boldsymbol{h} = f_{enc}(\boldsymbol{x}), \boldsymbol{h}_t = f_{enc}(\boldsymbol{x}_t). \tag{1}$$

Here, $\odot$ denotes the element-wise dot product operation, $f_{dec}$ and $f_{enc}$ denotes a decoder and an encoder, respectively, $\boldsymbol{x}_t$ is the input of a representative target class sample from which we extract the trigger pattern, and $\boldsymbol{m}$ is a mask vector whose values range from 0 to 1. The key to our strategy is to extract $\boldsymbol{x}_t$ and specify $\boldsymbol{m}$. Below we illustrate the details of our strategy.

### 3.2   EXTRACT TRIGGER PATTERN FROM BENIGN DATA

Inspired by the idea of the sample-specific attack strategy (Li et al., 2021c), we propose to extract the trigger pattern and inject it into a benign input in the hidden feature space. Specifically, we train an auto-encoder using the benign training dataset, where the learning of the encoder is additionally supervised by the classification object. Let $f_{enc}$, $f_{dec}$, and $f_{cls}$ denote the encoder, decoder, and classifier, respectively. The three modules work as follows:

$$\boldsymbol{h} = f_{enc}(\boldsymbol{x}), \quad \hat{\boldsymbol{x}} = f_{dec}(\boldsymbol{h}), \quad p(y|\boldsymbol{x}) = f_{cls}(\boldsymbol{h}), \tag{2}$$

and they are jointly trained on a reconstruction loss, $\mathcal{L}_{rec}$, and a classification loss, $\mathcal{L}_{cls}$, defined as follows:

$$\mathcal{L}_{rec} = \sum_{\boldsymbol{x} \in \mathcal{D}} ||\hat{\boldsymbol{x}} - \boldsymbol{x}||_2, \quad \mathcal{L}_{cls} = \sum_{(\boldsymbol{x}, y) \in \mathcal{D}} \ell_{ce}\left(y, p(y|\boldsymbol{x})\right), \tag{3}$$

where $\ell_{ce}$ denotes the cross-entropy-based loss function. During the training of these modules, we mine the representative sample $\boldsymbol{x}_t$ as described in the next section, and once the training done, we mine the mask $\boldsymbol{m}$.

#### 3.2.1   EXTRACT REPRESENTATIVE SAMPLE

According to the objective of our strategy, we should extract a pattern that occurs frequently in samples of the target class but rarely in samples of non-target classes as the trigger pattern. However, in numerous tasks, it is complicated to define the form of a pattern. Therefore, instead of extracting the trigger pattern directly, we first extract an input of the target class, denoted as $\boldsymbol{x}_t$, which should contain patterns that satisfy our requirement. Note that a pattern satisfying our requirement is a strong feature in the classification task to recognize the target class, and since deep models tend to adapt easy patterns first and then hard patterns (Arpit et al., 2017; Zhang et al., 2017), an input containing the trigger pattern should be recognized as the target class by the deep model in the early stage of the model training. Our solution for extracting $\boldsymbol{x}_t$ builds on this intuition.

Specifically, at the end of the **first training epoch** of $f_{enc}$ and $f_{cls}$, we feed all training samples of the target class to the encoder and the classifier, obtaining their prediction probabilities for the target class. We select the input with the highest prediction probability for the target class as the representative input, denoted as $\boldsymbol{x}_t$.

| Dataset | # Classes | #Train / #Test | $f_{vim}$ |
|---------|-----------|----------------|-----------|
| MNIST | 10 | 60,000/10,000 | 3 ConvBlocks, 2 fc |
| CIFAR10 | 10 | 50,000/10,000 | PreActResNet18 |
| GTSRB | 43 | 39,209/12,630 | PreActResNet18 |
| CelebA | 8 | 162,770/19,962 | ResNet18 |

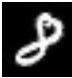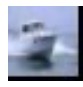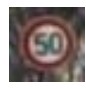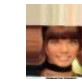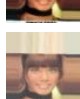

(a) Datasets and model architectures

(b) Poisoned data visualization

Figure 1: Data information. (a) Data statistic and classifier architectures. (b) Visualization of the poisoned images produced by our method compared to the corresponding benign images. The first row are benign images, and the second row are poisoned images.

### 3.2.2 MASK GENERATION

Intuitively, the encoded representation, $h_t$ of the selected input, $x_t$, should contain some redundant patterns (e.g., those used for input reconstruction) that do not meet our trigger pattern requirement. Directly merging $h_t$ with the encoded representation $h$ of a benign input $x$ may overwhelm the semantics of the benign input. With this consideration, we introduce a mask to reduce the redundant patterns in $h_t$ based on the intuition that removing the redundant patterns has less impact on the prediction of the labelling of $x_t$.

Specifically, we train a mask $m$ to remove the redundant patterns in $h_t$ with the following object:

$$\mathcal{L}_{mask} = \ell_{ce}\left(c_t, p(y|\tilde{h}_t)\right) + \alpha \sum_i^d m_i + \sum_{i=1}^d -m_i \log m_i, \text{with} \tag{4}$$
$$m = \sigma(\theta_m), \quad \tilde{h}_t = m \odot h_t, \quad p(y|\tilde{h}_t) = f_{cls}(\tilde{h}_t),$$

where $d$ denotes the dimension of $h_t$, $\theta_m$ denotes the trainable parameter to generate $m$, $\alpha$ denotes a scalar hyper-parameter, and $\sigma(\cdot)$ denotes the sigmoid activation function. The first term of the above learning object discourages the masking of patterns satisfying our trigger pattern requirement. The second term encourages masking as many elements of $h_t$ as possible. The last term encourages binarizing the mask. We optimize $\theta_m$ to minimize $\mathcal{L}_{mask}$ with the parameters of $f_{cls}$ fixed. In our implementation, we adjusted the value of $\alpha$ to mask 70%-80% of the elements of $h_t$.

## 4 EXPERIMENTS

### 4.1 EXPERIMENTAL SETTINGS

**Datasets & DNNs.** We performed experiments on four widely-used image datasets: MNIST (Le-Cun et al., 1998), CIFAR10 (Krizhevsky et al., 2009), GTSRB (Stallkamp et al., 2012), and CelebA (Liu et al., 2015). For implementing the model, we fellow most of the setting of WaNet (Nguyen & Tran, 2021) and considered a mixed of popular models: VGG11 and Pre-activation Resnet-18 (He et al., 2016) for CIFAR-10 and GTSRB and Resnet-18 for CelebA. For MNIST, we employed the same CNN model used by WaNet. Figure 1a shows some statistic information of these datasets.

**Attack Baselines.** We implemented three popular attack strategies as baselines: visible **BadNet** (Gu et al., 2017), which injects a checkerboard into the upper left corner of the input image; invisible **WaNet** (Nguyen & Tran, 2021), which used a small and smooth warping field to create poisoned images; and instance-specific **ISSBA** (Li et al., 2021c), which used an autoencoder to generate instance-specific backdoors. Note that we did not apply WaNet's noise mode because it needs to control the training process of the victim model. This is exactly what we want to avoid.

**Defense Algorithms.** We implemented five representative defense algorithms, including the model reconstruction-based algorithm **NAD** (Li et al., 2021a) and **DBD** (Huang et al., 2022b), the model diagnosis-based algorithm **STRIP** (Gao et al., 2019), the trigger synthesis based algorithm **Neural-Cleanse** (Wang et al., 2019), and the input preprocessing based algorithm **ShrinkPad** (Li et al., 2021b)For ShrinkPad, NAD, and DBD, we reported the Attack Success Rate (ASR) to evaluate

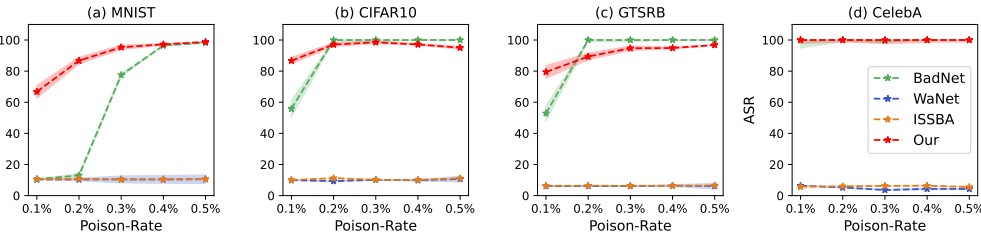

Figure 2: ASR × 100 by poison rate.

the performance of the attack. A stealthier attack against ShrinkPad, NAD, and DBD should achieve a higher value for ASR. For STRIP, we report the ratio of poisoned samples that escape STRIP's identification (i.e., False Accept Rate (**FAR**)) when **99%** of benign samples are not incorrectly recognized as poisoned samples, and the ratio of benign samples that are detected as poisoned samples (i.e., False Reject Rate (**FRR**)) when **99%** poisoned samples are correctly recognized by STRIP. A stealthier attack against STRIP should yield higher values for FAR and FRR. As for Neural-Cleanse, we report the Anomaly Index value to evaluate the stealthiness of the attack.

**Implementation Detail.** Following the practice of WaNet, we changed the input image size to $32{\times}32{\times}3$ for CIFAR10 and GTSRB, $28{\times}28$ for MNIST, and $64{\times}64{\times}3$ for CelebA. Normalization was applied to MNIST and CIFAR10. RandomCrop and RandomHorizontalFlip were applied on CIFAR10. We applied CNN based auto-encoders to implement $f_{enc}$ and $f_{dec}$. The classifier $f_{cls}$ was implemented with a MLP. The target class was set to 2 for all the datasets. Early stop was applied to select the model based on performance on benign data. **Label-smoothing** (Szegedy et al., 2016) was applied as a regularization during the training of the victim model, with $\hat{\boldsymbol{y}}_i = 0.8\boldsymbol{y}_i + \frac{0.2}{C}$, where $C$ denotes the class number and $\boldsymbol{y}_i$ denotes the i-th dimension value of $\boldsymbol{y}$, which is a one-hot label representation. Label smoothing is a popular technique to address label noise and according to our experience, it is highly effective to defend poison-label backdoor attacks. Momentum optimizer with 0.01 initial learning rate was applied for training the model. $\alpha$ was set to 5 on MNIST, and 8 on the other three datasets. Please refer to the appendix in the end of this manuscript and the attached **source code** in the zip file for more details of the implementation.

## 4.2 EFFICIENCY STUDY

Figure 2 shows the attack performance of different strategies when poisoning different rates of training data. From the figure we can see: **1)** WaNet and ISSBA cannot achieve a success attack when poisoning 0.1% to 0.5% of training data. Actually, according to our experiments, when label smoothing is applied, WaNet cannot achieve an almost 100% ASR until the poison rate reaches 5% on MNIST and CelebA, and 10% on CIFAR10 and CelebA. ISSBA cannot achieve a success attack even if the poison rate reaches 10% when label smoothing is applied. **2)** Our strategy performs much better than all the compared baselines when only 0.1% (i.e., **40-60** samples) of training data is poisoned on MNIST, CIFAR10, and GTSRB, and performs similar to BadNet on CelebA. **3)** On the black-and-white image-based MNIST dataset, our strategy consistently outperforms BadNet, while on the rest color image-based datasets, BadNet performs similar or somehow better than our strategy when the poison rate is more than 0.1%.

From these observations, we draw a rough conclusion about the attack efficiency of the tested strategies. On the black-and-white image based dataset, the order of attack efficiency is: Our > BadNet ≫ WaNet > ISSBA. On the color image based datasets, the order of attack efficiency is: BadNet ≈ Our ≫ WaNet > ISSBA. We believe that the reason BadNet is much more efficient on color image-based datasets is that the black-and-white checkerboard trigger used by BadNet looks too conspicuous in the color image to be fitted by the model.

Of course, the above conclusion may be not so rigorous, as many factors, such as the size of the trigger, where the trigger is inserted, and the brightness of the trigger, affect attack performance. Changing these factors may increase the efficiency of the baselines. We fellow the public setting (Li et al., 2022b) in our experiments and did not change the setting since it was not the focus of this

Table 1: BA × 100 and ASR × 100 without any defense.

| Dataset | BA | | | | ASR | | |
|---|---|---|---|---|---|---|---|
| | w/o attack | BadNet | WaNet | Our | BadNet | WaNet | Our |
| MNIST | 99.21 | 99.18 | 99.12 | 99.19 | 100 | 99.76 | 100 |
| CIFAR10 | 94.12 | 93.44 | 93.01 | 93.44 | 99.82 | 99.99 | 99.66 |
| GTSRB | 97.27 | 96.83 | 97.01 | 97.95 | 100 | 97.37 | 99.03 |
| CelebA | 79.38 | 79.10 | 79.10 | 79.20 | 100 | 100 | 99.99 |

Table 2: Attack Performance against (a) STRIP, and (b) Neural-Cleanse. (a) FAR × 100 when FRR = 1% and FRR × 100 when FAR = 1% against STRIP. (b) The asterisk means that Neural-Cleanse cannot correctly recognize the attacker-defined target class even if it correctly identifies the model is attacked.

(a) FAR × 100 and FRR × 100 against STRIP

| Dataset | FAR | | | FRR | | |
|---|---|---|---|---|---|---|
| | BadNet | WaNet | Our | BadNet | WaNet | Our |
| MNIST | 0 | 0.24 | **51.80** | 0 | 0.36 | **13.31** |
| CIFAR10 | 92.70 | 96.23 | **100** | 81.32 | 83.33 | **87.30** |
| GTSRB | 0 | 0.22 | **60.14** | 0 | 0.15 | **95.05** |
| CelebA | 33.53 | 6.55 | **51.42** | 19.90 | 7.50 | **63.40** |

(b) Anomaly Index against Neural-Cleanse

| Dataset | BadNet | WaNet | Our |
|---|---|---|---|
| MNIST | 5.17 | 1.14 | **0.79** |
| CIFAR10 | 5.75 | 8.92 | **1.65**[*] |
| GTSRB | 13.19 | 12.11 | **4.13**[*] |
| CelebA | 0.71[*] | 0.85[*] | 0.85[*] |

work and the results were enough to support our declaim: our proposed strategy is very efficient in attacking.

### 4.3 STEALTHINESS STUDY

In this section, we study stealthiness of attacks using different strategies, when they can achieve a success attack without any defense. To perform the study, we set the poison rate to 5% for all the strategies and remove label smoothing during victim model training. This is because label smoothing will significantly degrade the attack efficiency of WaNet and ISSBA. It has to set a high and unacceptable value of poison rate for them to achieve a successful attack, which we want to avoid. Besides, without further illustration, we did not report the results of ISSBA because it cannot achieve acceptable attack performance in this setting even without any defense, and further increasing the poison rate will make the attack inapplicable in real situations.

**Backdoor Visibility & Performance without Defense.** Figure 1b shows the poisoned images generated by our method compared to their corresponding benign ones. As can be seen from this figure, the clarity of the images is somewhat reduced by the injection of the trigger pattern generated by our method. However, the poisoned images are still meaningful and it is hard to tell what the trigger pattern is. In addition, as can be seen, the presentation of the trigger pattern is input independent in the input space. It is hard to find the commodity of the trigger pattern in different poisoned inputs in the input space. This meets up with the design theory of ISSBA. Table 1 shows the attack performance without any defense. From the table it can be seen that all tested attack strategies can achieve similar BAs as the intact model (w/o attack) that is not attacked. Moreover, they all achieve an almost 100% ASR, which proves their effectiveness when no defense system is used.

**Resistance to STRIP.** Table 2a shows the attack performance against STRIP. From the table, we can see that STRIP performs quite effectively in defending backdoor attacks, especially for those using the BadNet and WaNet strategy, and that attacks using our strategy achieve the best performance against STRIP in most cases. As a rough conclusion, we can say that in terms of stealthiness against STRIP: Our > WaNet > BadNet. Here we give the reason why our strategy performs well against STRIP. STRIP builds on the phenomenon that label prediction is quite robust to perturbations in the input when the trigger pattern occurs in the input, and it performs defense by removing samples with small deviations in label prediction when the input is perturbed. In our strategy, the trigger pattern is from the target class, so the deviations in label prediction are also small

Table 3: Attack performance against (a) NAD, (b) ShrinkPad, and (c) DBD. We refer the reader to Appendix C for results on BA.

<table>
<tr><td colspan="4">(a) ASR against NAD</td><td colspan="4">(b) ASR against ShrinkPad</td><td colspan="4">(c) ASR against DBD</td></tr>
<tr><td>Dataset</td><td>BadNet</td><td>WaNet</td><td>Our</td><td>Dataset</td><td>BadNet</td><td>WaNet</td><td>Our</td><td>Dataset</td><td>BadNet</td><td>WaNet</td><td>Our</td></tr>
<tr><td>MNIST</td><td>10.84</td><td>10.55</td><td>**33.01**</td><td>MNIST</td><td>25.40</td><td>**99.72**</td><td>88.69</td><td>MNIST</td><td>100</td><td>88.75</td><td>**100**</td></tr>
<tr><td>CIFAR10</td><td>20.08</td><td>87.46</td><td>**94.93**</td><td>CIFAR10</td><td>34.75</td><td>62.29</td><td>**87.74**</td><td>CIFAR10</td><td>1.64</td><td>1.42</td><td>**99.99**</td></tr>
<tr><td>GTSRB</td><td>58.50</td><td>56.90</td><td>**89.84**</td><td>GTSRB</td><td>69.67</td><td>39.69</td><td>**88.05**</td><td>GTSRB</td><td>3.73</td><td>0.0</td><td>**99.81**</td></tr>
<tr><td>CelebA</td><td>12.43</td><td>19.70</td><td>**37.54**</td><td>CelebA</td><td>77.22</td><td>81.43</td><td>**96.39**</td><td>CelebA</td><td>0.0</td><td>0.0</td><td>**99.17**</td></tr>
</table>

for many benign samples of the target class. This makes it difficult to set a threshold to distinguish between poisoned and benign samples, leading to the above observations.

**Resistance to Neural-Cleanse.** Table 2b shows the performance of the attack against Neural-Cleanse. The asterisk indicates that Neural-Cleanse cannot correctly identify the attacker-defined target class $c_t$ using the pattern norms, even if it had recognized the attacked model. This also indicates that the attack is very stealthy towards Neural-Cleanse and in this case it may not compare the stealthiness of attacks based on their anomaly index values. From the table, it can be seen that the attack performed with our strategy achieves the smallest value of Anomaly Index on MNIST. For the other three datasets, Neural-Cleanse cannot identify the attacker-defined target class of our attacks. This proves the great stealthiness of our strategy against Neural-Cleanse. In comparison, the attacks using BadNet and WaNet strategies achieve larger Anomaly Index values and their attacker-defined target classes can be identified by Neural-Cleanse on all datasets except CelebA. As an explanation for this phenomenon, we hypothesize that the inconspicuousness of our strategy against Neural-Cleanse stems from the practice of poisoning data in the hidden space as ISSBA, which causes the trigger pattern to differ depending on the input in the input space.

**Resistance to NAD.** Table 3a shows the attack performance against NAD. NAD is sensitive to its hyper-parameters. We selected the model with BA during the application of NAD. The optimizer and learning rate were adjusted so that BA does not drop more than 10% (otherwise the defense is useless as it degrades the benign performance too much). From the table we can see that attacks performed with our strategy achieve much higher ASRs than the baselines, and we can roughly conclude that in terms of stealthiness against NAD: Our > WaNet > BadNet. Here we give the reason why our strategy performs well against NAD. Basically, NAD performs defense by preventing the activation of backdoor-specific neurons through attention distillation. Our trigger pattern frequently occurs in benign samples of the target class, so it is difficult to prevent the activation of our trigger pattern. Otherwise, BA will drop significantly.

**Resistance to ShrinkPad.** Table 3b shows the attack performance against ShrinkPad. Similarly, when using ShrinkPad, we selected the model with BA. The padding size was adjusted so that BA did not drop more than 10%. From the table, we see that the attacks performed with our strategy achieve significantly higher ASRs than the baselines on all datasets except MNIST, where WaNet performs best. WaNet performs better than BadNet in most cases. These observations are in line with our expectations since BadNet uses a patch-based trigger pattern, while WaNet and our strategy have the trigger pattern scattered throughout the input space. This makes attacks with WaNet and our strategy robust to input noise. Compared to our strategy, WaNet merges the trigger pattern directly with the input in the input space, while our strategy performs trigger pattern injection in the hidden space. This is likely the reason why our strategy performs better than WaNet against ShrinkPad. All in all, we can roughly conclude that in terms of stealthiness against ShrinkPad: Our > WaNet > BadNet.

**Resistance to DBD.** Table 3c shows the attack performance against DBD. From the table, we can first see that DBD is very effective in defending attacks applied to the three color image datasets using the BadNet and WaNet strategies. However, it does not perform as well on MNIST and cannot defend against attacks performed with our proposed strategy. Our explanation for the first phenomenon is that all strategies work very efficiently on MNIST. Some poisoned samples were already fitted well at the second training stage. These are treated as high credible samples and

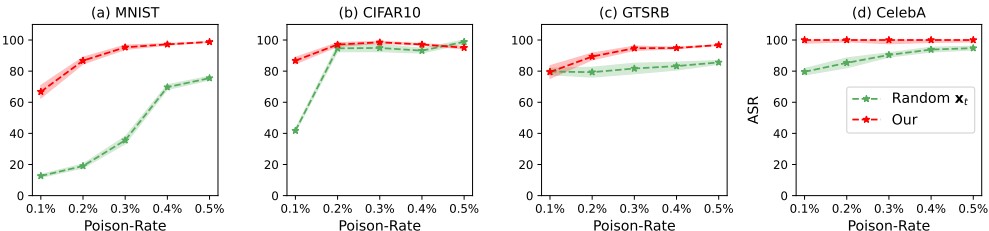

Figure 3: Attack performance by poison rate when using different strategies to select $\boldsymbol{x}_t$.

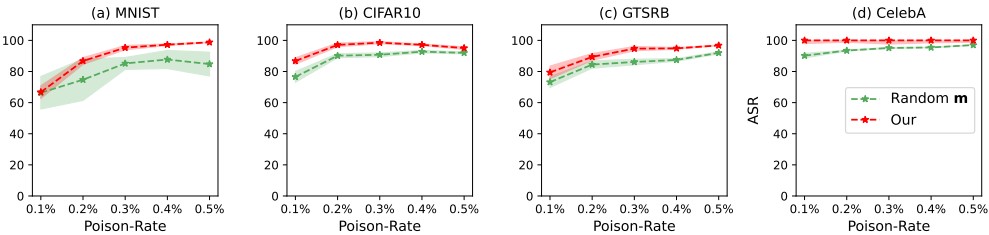

Figure 4: Attack performance by poison rate when using different methods to set $\boldsymbol{m}$.

selected for fine-tuning the model in the third stage, resulting in a high value of ASR due to the high attack efficiency. Such an explanation also applies to the phenomenon that occurs in attacks using our strategy, considering that our strategy works very efficiently on all datasets.

### 4.4 ABLATION STUDY

**Influence of the selection of $\boldsymbol{x}_t$.** Here we explore the advantage of our proposed method in selecting the representative sample, $\boldsymbol{x}_t$, as defined in section 3.2.1. For the study, we drew a random sample of the target class from the training dataset as $\boldsymbol{x}_t$ and compared its performance with the proposed method defined in section 3.2.1. Figure 3 shows the comparison results. There, a dot line associated with a legend "Random" corresponds to the performance of the attack with the random strategy. As can be seen from the figure, our proposed strategy generally performs better than the random strategy, especially when the poison rate is low. When the poison rate reaches 0.5%, even the random strategy is quite effective. This proves the advantage and robustness of our proposed strategy, which uses patterns of the target class defined by the attacker as trigger patterns.

**Influence of the setting of $\boldsymbol{m}$.** Here, we study the effectiveness of our proposed method for defining the mask $\boldsymbol{m}$. To conduct the study, we randomly select $n$ elements of $\boldsymbol{m}$ and set their values to 1, while the other elements are set to 0. The value of $n$ was set to the sum of the mask generated by our method. Figure 4 shows the comparing results with our proposed method. Surprisingly, given a randomly generated mask $m$ (the number of masked elements is the same as in our proposed method), our strategy can still achieve great attack efficiency. One possible explanation for this phenomenon is that there are many patterns in $\boldsymbol{h}_t$ that satisfy the requirement of our strategy. Thus, it is very likely that we can obtain suitable patterns using a random mask.

## 5 CONCLUSION

In this paper, we present a new strategy for defining the trigger pattern of poison label backdoor attacks. Instead of using a rarely occurring pattern in benign data as the trigger pattern, patterns that occur frequently in benign samples of the predefined target class but rarely in samples of non-target classes are extracted as the trigger pattern. Accordingly, we present an effective technique to extract the trigger pattern and insert it into benign inputs for data poisoning. Experimental studies on several benchmark tasks show that our proposed strategy can significantly improve the efficiency and stealthiness of the attack.

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

Table 4: Architectures and hyper-parameters of $f_{enc}$ and $f_{dec}$.

| | CIFAR10 | GTSRB | MINIST | CelebA |
|---|---|---|---|---|
| $f_{enc}$ | Conv2D(3, 12, 4, 2, 1) | Conv2D(3, 12, 4, 2, 1)) | Conv2D(1, 12, 4, 2, 1) | Conv2D(3, 12, 4, 2, 1) |
| | Conv2D(12, 24, 4, 2, 1) | Conv2D(12, 24, 4, 2, 1) | Conv2D(12, 24, 4, 2, 1) | Conv2D(12, 24, 4, 2, 1) |
| | Conv2D(24, 48, 4, 2, 1) | Conv2D(24, 48, 4, 2, 1) | Conv2D(24, 48, 4, 2, 0) | Conv2D(24, 48, 4, 2, 1) |
| | | | | Conv2D(48, 96, 4, 2, 1) |
| $f_{dec}$ | Conv2DT(48, 24, 4, 2, 1) | Conv2DT(48, 24, 4, 2, 1) | Conv2DT(48, 24, 4, 2, 0) | Conv2DT(96, 48, 4, 2, 1) |
| | Conv2DT(24, 12, 4, 2, 1) | Conv2DT(24, 12, 4, 2, 1) | Conv2DT(24, 12, 4, 2, 0) | Conv2DT(48, 24, 4, 2, 1) |
| | Conv2DT(12, 3, 4, 2, 1) | Conv2DT(12, 3, 4, 2, 1) | Conv2DT(12, 1, 4, 2, 1) | Conv2DT(24, 12, 4, 2, 1) |
| | | | | Conv2DT(12, 3, 4, 2, 1) |
| $f_{cls}$ | Linear(768, 256) | Linear(768, 256) | Linear(192, 128) | Linear(1536, 1024) |
| | Linear(256, 128) | Linear(256, 128) | Linear(128, 64) | Linear(1024, 128) |
| | Linear(128, 10) | Linear(128, 43) | Linear(64, 10) | Linear(128, 8) |

Table 5: BA $\times$ 100 and ASR $\times$ 100 against NAD.

| Dataset | BA | | | ASR | | |
|---|---|---|---|---|---|---|
| | BadNet | WaNet | Our | BadNet | WaNet | Our |
| MNIST | 97.14 | 97.73 | 97.32 | 10.84 | 10.55 | **33.01** |
| CIFAR10 | 89.74 | 89.94 | 89.47 | 20.08 | 87.46 | **94.93** |
| GTSRB | 96.01 | 97.68 | 96.41 | 58.50 | 56.90 | **89.84** |
| CelebA | 77.39 | 76.95 | 76.71 | 12.43 | 19.70 | **37.54** |

## A  IMPLEMENTATION DETAIL

**Architecture of $f_{enc}$ and $f_{dec}$**  The auto-encoder architectures and hype-parameters for four data sets are listed in Table 4. The encoder has several layers of CNN with relu function as the activate function. The decoder has two branches. One is a regressive decoder (Reg Decoder) which consist of several transposed CNN layer. relu is adopted as the activation function in each layer except for the last layer which uses the sigmoid function to scale the output value in between (0, 1). The other branch is the classification decoder (Cls Decoder) which consists of several linear layers with relu function in the middle.

## B  MASK VISUALIZATION

Figure 5 shows the decoded representative input and their masked counterparts. From the figure, we can see that it inclines to extract the outline the representative input as the trigger pattern using the mask.

## C  STEALTHINESS STUDY

Table 5, 6, and 7 show BAs and ASRs of attacks against NAD, ShrinkPad, and DBD, respectively.

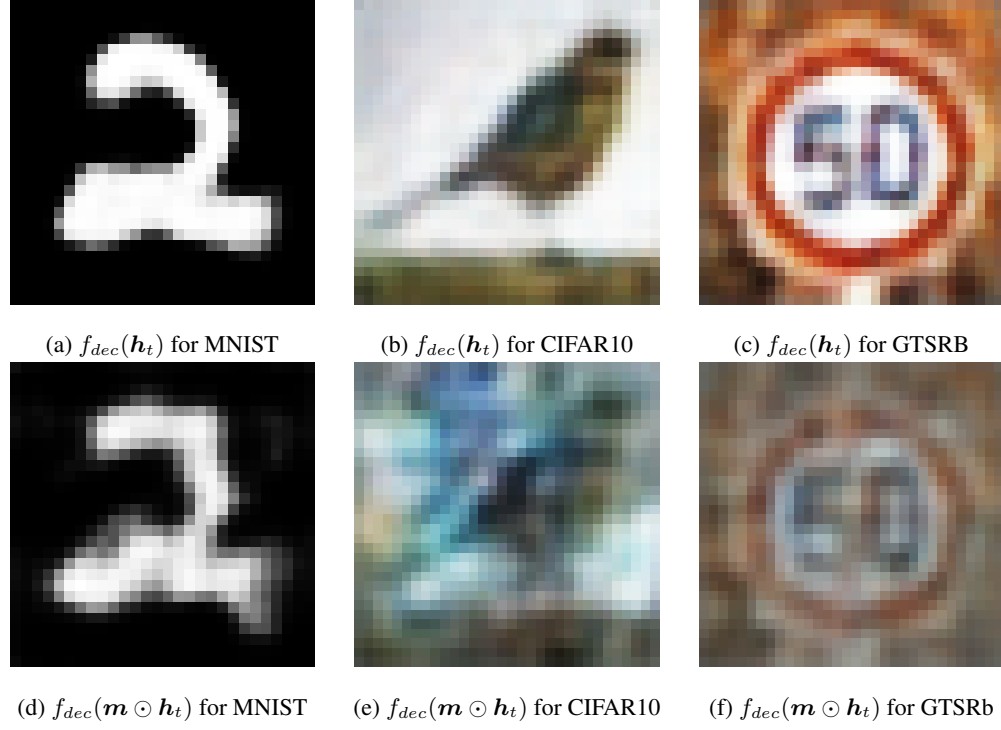

(a) $f_{dec}(\boldsymbol{h}_t)$ for MNIST     (b) $f_{dec}(\boldsymbol{h}_t)$ for CIFAR10     (c) $f_{dec}(\boldsymbol{h}_t)$ for GTSRB

(d) $f_{dec}(\boldsymbol{m} \odot \boldsymbol{h}_t)$ for MNIST     (e) $f_{dec}(\boldsymbol{m} \odot \boldsymbol{h}_t)$ for CIFAR10     (f) $f_{dec}(\boldsymbol{m} \odot \boldsymbol{h}_t)$ for GTSRb

Figure 5: Visualization of the decoded representative inputs and their masked counterparts.

Table 6: BA $\times$ 100 and ASR $\times$ 100 against ShrinkPad.

| Dataset | BA | | | ASR | | |
|---|---|---|---|---|---|---|
| | **BadNet** | **WaNet** | **Our** | **BadNet** | **WaNet** | **Our** |
| MNIST | 97.35 | 99.12 | 97.15 | 25.40 | **99.72** | 88.69 |
| CIFAR10 | 89.94 | 89.42 | 86.01 | 34.75 | 62.29 | **87.74** |
| GTSRB | 93.57 | 94.68 | 97.35 | 69.67 | 39.69 | **88.05** |
| CelebA | 75.92 | 77.90 | 78.20 | 77.22 | 81.43 | **96.39** |

Table 7: BA $\times$ 100 and ASR $\times$ 100 against DBD.

| Dataset | BA | | | ASR | | |
|---|---|---|---|---|---|---|
| | **BadNet** | **WaNet** | **Our** | **BadNet** | **WaNet** | **Our** |
| MNIST | 97.40 | 97.76 | 97.27 | 99.15 | 97.27 | **100** |
| CIFAR10 | 86.51 | 84.58 | 87.39 | 1.64 | 1.42 | **99.99** |
| GTSRB | 93.38 | 93.02 | 93.40 | 3.73 | 0.0 | **99.81** |
| CelebA | 65.27 | 65.99 | 65.87 | 0.0 | 0.0 | **99.17** |

