# OpenReview forum: "Efficient and Stealthy Backdoor Attack Triggers are Close at Hand"
_ICLR.cc/2023/Conference — Submitted to ICLR 2023_

### Official Review · Reviewer_Q7gZ · 2022-10-19

**Confidence:** 4
**Correctness:** 3
**Technical Novelty And Significance:** 2
**Empirical Novelty And Significance:** 2
**Recommendation:** 3

**Clarity, Quality, Novelty And Reproducibility:**

## Clarity

This paper is mostly well written and easy to follow. The introduction provides sufficient background and intriguing motivation to the problem that this paper aims to address. The proposed attack method is mostly easy to understand.

There is one part not clear from the description. In section 3.2.1, this paper states "we select the input with the highest prediction probability for the target class as the representative input." Does it mean only one single sample from the target class is used for the attack? If this is true, it is unclear why the features from one sample is sufficient to achieve high attack success rate? It would be better to provide more details and explanations.

Besides, this paper does not provide a threat model for the proposed attack. What knowledge is needed for the attacker and what capability does the attacker have?


## Novelty

This paper proposes a backdoor attack leveraging an encoder-decoder structure to combine features from the target-class sample and victim samples. The backdoor trigger is sample-specific. This is very similar to an existing attack, Invisible backdoor attack [1], which also makes use an auto-encoder to generate sample-specific backdoor patterns. This paper states the proposed attack is "inspired by the idea of the sample-specific attack strategy [1]". However, there is discussion regrading the difference between the proposed one and [1]. Besides, the existing attack [1] is not compared with in the evaluation.

The trigger injection method of this paper is using a mask to combine features of samples from victim and target classes. Such an analytic technique has already been studied in existing work [2], which aims to differentiate internal features from victim and target classes using an optimized mask. This method is able to detect many complex backdoor attacks. It shall be discussed and evaluated in the paper as it shares similarities with the proposed attack.


## Quality

The evaluation includes a few defense methods. The results in Table 2(b) show NC can detect 3 out of 4 backdoored models by the proposed attack. The paper sees such results promising as NC cannot identify the correct target label. However, it is not a convincing argument. Backdoor scanners such as NC only aim to determine whether a model is backdoored or not. If it is backdoored, users then can just discard the model and do not use it. It does not matter whether the correct target label can be identified or not. This also echoes the previous concern that this paper does not provide an explicit threat model.

In addition, there are many more recent state-of-the-art defenses in backdoor scanning [2] and backdoor removal [3-5]. This paper shall also evaluate against these new defenses.


## Reproducibility

The submission provides the implementation of the proposed method in the supplementary material, which makes the reproduction easy and achievable.


* [1] Yuezun Li, et al. "Invisible backdoor attack with sample-specific triggers." ICCV 2021
* [2] Liu, Yingqi, et al. "Complex Backdoor Detection by Symmetric Feature Differencing." CVPR 2022.
* [3] Wu, Dongxian, and Yisen Wang. "Adversarial neuron pruning purifies backdoored deep models." NeurIPS 2021.
* [4] Tao, Guanhong, et al. "Model orthogonalization: Class distance hardening in neural networks for better security." IEEE S&P 2022.
* [5] Zeng, Yi, et al. "Adversarial Unlearning of Backdoors via Implicit Hypergradient." ICLR 2022.

**Strength And Weaknesses:**

Strength

+ Interesting perspective of using target-class features as the trigger
+ Well written

Weaknesses

- No threat model
- Limited novelty
- Unconvincing results against defenses
- No comparison with a closely related backdoor attack
- Missing evaluation on state-of-the-art defenses

**Summary Of The Paper:**

This paper proposes a backdoor attack using features from the target class. Existing backdoor attacks usually use some pattern that does not relate to the target label, which could be identified by defense methods. This paper leverages the benign features from the target class as the trigger pattern to inject backdoor behaviors. Specifically, it selects a sample from the target class based on the prediction probability. It then utilizes an encoder-decoder structure to extract the feature representation of the target-class sample, which is combined with the feature representation of samples from other classes as trigger-injected samples (after decoded by the decoder). The number of the added features from the target-class sample is minimized using the classification loss. The evaluation is conducted on four datasets and a few model architectures. The results show the proposed attack can achieve high attack success rate and has better resilience to defenses compared to a few existing backdoor attacks.

**Summary Of The Review:**

This paper is interesting and mostly well written. The novelty and the evaluation are however limited.

---

> ### Author Response · Authors · 2022-11-17
> **Thank you for your review. Here are our answers.**
>
> Thank you for your thorough review. We hope that our response will address your concerns.
>
> 1.**Regarding the concern about the statement “we select the input with the highest prediction probability for the target class as the representative input” proposed in the Clarity part:** \
> This statement describes the extraction of the representative sample but not the data poisoning process. In our method, we first extract the trigger pattern from a single representative sample. After that, we insert the trigger pattern to a few samples (the number depends on the data poison rate) to perform data poisoning and consequently, the backdoor attack.
>
> 2.**Regarding the concern about the threat model:** \
> In the submitted manuscript, we put the statement about the threat model of our method in the first paragraph of section 2.1. Specifically, we assume that attackers are allowed to poison some training data, whereas they have no information on or change other training components (e.g., training loss, training schedule, and model structure). In the inference process, attackers can and only can query the trained model with any input. We have added a highlight title for this part and updated the manuscript.
>
> 3.**Regarding the concern about the comparison with ISSBA:** \
> We have added a description of ISSBA in the “Previous Poison-Label Backdoor Attacks” paragraph of section 2.1. The key difference between our method and ISSBA lies in the trigger pattern design. ISSBA uses a human-designed string as the trigger pattern. Such a pattern does not exist in benign data. While our method extracts a representative pattern of the target class as the trigger pattern. Such a pattern already exists in benign data. Though both ISSBA and our method use an auto-encoder to perform data poisoning, the used trigger patterns are greatly different.
>
> As for the empirical comparison, the experimental study mainly contains two parts, i.e., the attack efficiency study（in **section 4.2**）and the attack stealthiness study (in **section 4.3**). In the attack efficiency study, we have compared the performance of our method with that of ISSBA. The experimental results (in **Figure 2**) reveal that the attack efficiency of ISSBA is far lower than our method.
>
> We did not compare with ISSBA in the attack stealthiness study because ISSBA requires poisoning too much training data and can be easily defended by label-smoothing, which is a very basic defense method. According to our experiments, ISSBA has to poison about 10% of training data to achieve a successful attack (almost 100% attack success rate) without any defense applied. This is consistent with the reported value in the corresponding paper of ISSBA. And when label-smoothing is applied, the required data poison rate reaches 20%. Such high data poison rates make ISSBA easy to be noticed by the developer during data analysis. After all, it is doubtful that 10% or more of samples are wrongly labeled as a consistent class.
>
> 4.**Regarding the concern about evaluating our method against more defense methods:** \
> We recognize that there are a huge number of defence methods being proposed in recent years. However, we do not think every defense method deserves a try. We have tested the reconstruction-based algorithms NAD and DBD, the model diagnosis-based algorithm STRIP , the trigger synthesis-based algorithm Neural-Cleanse, and the input preprocessing-based algorithm ShrinkPad.  In addition, thanks to the suggestion of Reviewer 1, we have tested a frequency-based defense method and the experimental results show that our method can successfully escape the defense of the frequency-based defense method (you may refer to our response to the first review for more detail). These methods are typical for each defense direction. Results on these methods can be migrated to other methods in the same direction.

---

> > ### Comment · Reviewer_Q7gZ · 2022-11-17
> > **Thanks for the Rebuttal**
> >
> > Thank the authors for responding to my comments.
> >
> > The trigger design difference between ISSBA and the proposed attack seems minor. This does not constitute a sufficient contribution. The rebuttal shows that ISSBA can be easily defended by label-smoothing. However, such a simple defense can be easily evaded with an adaptive attack (knowing the usage of label-smoothing during attack). This is not a strong argument. I would like to see the comparison on more defense techniques.
> >
> > I agree with the authors that there are many defense methods and we cannot evaluate each and every one of them. I would suggest to use more recent and advanced ones. NC was published in 2019, which is three years ago. There are many following-up works that outperform NC. We shall use more recent state-of-the-art methods instead of old and weak baselines.
> >
> > With the above concerns unaddressed, I will keep my score.

---

> > > ### Author Response · Authors · 2022-11-18
> > > **Thanks for the Response**
> > >
> > > Thank the review for responding to the rebuttal.
> > >
> > > 1.**Improving the stealthiness against label-smoothing may not be so easy:** \
> > > Just for an academic discussion. We are curious about how to improve the stealthiness of ISSBA without changing the trigger pattern when the attackers know that label-smoothing will be applied for training the victim model. We think this is not so easy. Of course, if the trigger pattern can be changed, this may be achieved by searching for a more effective representative string for ISSBA. However, this is highly time-consuming. It has to train an auto-encoder and a predefined model (architecture may be different from that of the victim model) for every change of the trigger pattern. Moreover, the effectiveness of such a method is also limited. If only the trigger pattern hardly occurs in benign training data, it needs to poison quite a few samples to achieve the attack, resulting in lower attack efficiency.
> > >
> > > 2.**What defense methods deserve to try:** \
> > > Thanks for your suggestions for the defense methods [2-5]. Yes, these methods are newer than some of the tested methods. However, we do not see their potential advance over the tested methods against our method. For example, in [2], it determines if the features of an inverted trigger are the **natural features** distinguishing the victim and target classes. If
> > > so, the model is clean. Otherwise, it is considered trojaned. From the principle of the method, it cannot defend our attack method since the trigger patterns used in our work commonly occur in benign target class data and are natural features. Yes, introducing the defense may make our baseline look newer but it would not introduce more information.
> > > We do not exclude testing on newer methods if only the methods lie in a new defense direction.

---

### Official Review · Reviewer_ns26 · 2022-10-20

**Confidence:** 4
**Correctness:** 3
**Technical Novelty And Significance:** 2
**Empirical Novelty And Significance:** 2
**Recommendation:** 3

**Clarity, Quality, Novelty And Reproducibility:**

### Novelty
* The approach is very similar to ISSBA, which also considers dirty-label dataset poisoning, except this paper adds much larger and visible perturbations to the images to allow for lower poison rates.

### Quality
* This paper is generally clearly written, with a few issues below:
	* Many of the citations are missing parentheses.
	* Figure 2: why can the bounds on ASR for CelebA exceed 100%?
* It is strange that the paper did not mention the details of the method in the Introduction section.
* As this is a data poisoning attack, it is important to discuss whether poisoned data are transferrable across model architectures and training configurations.

### Reproducibility issues
* It is not clear how the mask is binarized. The paper only mentions the last term in eq. (4) can help with this. Binary masking typically requires gradient estimators for differentiable on/off decisions, iterative pruning, Gumbel-softmax hardening tricks, or even resort to RL algorithms. In general, you would want to "ease in" binary decisions, otherwise the result would depend heavily on initialization. It would also be nice to have some figures showing the masks and respective images.
* The paper mentions that $\alpha$ is adjusted to mask 70-80% of all elements in $\mathbf{h}_t$. The difficulty of this tuning step is not made clear.
* The training configurations for the auto-encoder is not provided in either the main text or the appendix.
* The defense hyperparameters are also missing.


**Strength And Weaknesses:**

### Strengths
* It appears to be quite effective with low poison rates, and can resist recent defenses.
* The idea of masking the encoded representation is somewhat interesting.

### Weaknesses
* The added perturbation is very visible and the labels are also changed, and can be easily detected by human, as shown in Fig. 1(b).
* Compared to the latest clean-label data poisoning methods (e.g. Sleeper Agent [1]), the proposed method further requires label-flipping and similar perturbation scale. It may be more practical for the attackers to consider clean-label variants.
* Table 2(b): the anomaly index of GTSRB seems to far exceed the limit (2) of Neural-Cleanse. WaNet also did not report such a high anomaly for GTSRB.
* The method may not be able to defend against MNTD [2].

### References
[1]: Souri H, et al. Sleeper agent: Scalable hidden trigger backdoors for neural networks trained from scratch, ICML workshop 2022.

[2]: Xu X, et al. Detecting AI trojans using meta neural analysis, S&P 2021.

**Summary Of The Paper:**

This paper proposes to find salient features of images of the target class in the encoded latent space with an auto-encoder, perturb the source image in its encoded representations with a learned mask, and decode the representation to form poisoned data for dirty-label data poisoning. Mask learning is performed with a presumably gradient-based optimization to minimize a combined loss. Experiments show that it is effective under low poison rates (0.1%) and can resist recent defenses.


**Summary Of The Review:**

The paper's method is somewhat novel, but not very well motivated. The proposed method is not stealthy enough, and "dirty-label" has greater restrictions in applicability and easier to detect. It lacks discussion on transferability as a data poisoning attack. It also has many reproducibility issues.

---

> ### Author Response · Authors · 2022-11-16
> **Thank you for your thorough review. Here are our answers.**
>
> Thank you for your thorough review. We hope that our response will address your concerns.
>
> 1.**Regarding the concern about the stealthiness of our attack against human inspection:** \
> We recognize the reviewer’s comment that our method requires label-flipping, making the attack easier to be detected by the developers through data analysis. This is why we address the efficiency of attacks, and we think the stealthiness problem against human inspection can be addressed by reducing the data poison rate. In practice, the developer usually samples a subset of the training data set for data analysis. With a 0.5% data poison rate and a 1000 sample size, it will only occur 5 poisoned samples in expectation. In such a case, it is hard for the developer to say that the data set has been poisoned rather than only containing a little label noise.
>
> 2.**Regarding the concern about the results of Neural-Cleanse:** \
> Yes, the reported performance of attackers against Neural-Cleanse is somehow different from that in the published paper of WaNet. We think this is because some settings of the experiment are different. Specifically,
> - To avoid controlling the training process of the victim model, we did not apply the noise mode of WaNet. This will result in an increase in the Anomaly-Index value. This is consistent with the reported results of the published paper of WaNet.
> - The data poison rate was set to 0.1 in WaNet while 0.05 in our experiments with the consideration of stealthiness against human inspection.
> - The target class was set to 0 in WaNet while 2 in our experiments.
>
> 3.**Regarding the concern about the binarization of the mask:** \
> We do not require the mask to fully satisfy the binarization constraint. We just encourage that through minimizing the entropy of every element of the mask: $- \mathbf{m}_i * \log \mathbf{m}_i – (1-\mathbf{m}_i)*\log(1-\mathbf{m}_i)$ as defined in Eq. (4).
>
> As for the visualization of the mask, we appreciate the suggestion of the review. However, we notice that it is meaningless to directly visualize the mask since it is applied to the hidden space and is somehow hard to understand for human beings. As an alternative, we visualize the decoded output of the masked representative sample, i.e., $f_{dec}(\mathbf{h}_t)$. We show the results in Appendix B of the updated manuscript.
>
> 4.**Regarding the concern about the tuning of $\alpha$:**  \
> We only tuned $\alpha$ to adjust the mask ratio. This makes the tuning of $\alpha$ straightforward: increasing $\alpha$ will increase the mask ratio. In addition, the computational cost for training the mask is just linear to the mask size. 10 seconds or so is enough on a GenuineIntel CPU to train the mask. During our implementation, we increased or decreased the value of $\alpha$ by 1 at every tuning step depending on the resulting mask ratio.
>
> 5.**Regarding the concern about the implementation detail of auto-encoder and defense methods:** \
> Most of the defense methods were implemented using the public source code with most of the settings not changed. In addition, we have uploaded the source code of our implementation as an appendix along with the submission. This is why we did not talk much about the implementation of the auto-encoder and the defense methods.

---

### Official Review · Reviewer_xZDL · 2022-10-23

**Confidence:** 4
**Correctness:** 3
**Technical Novelty And Significance:** 2
**Empirical Novelty And Significance:** 2
**Recommendation:** 5

**Clarity, Quality, Novelty And Reproducibility:**

I think this paper is easily achievable, but there is still further improvement in quality and clarity. In addition, the current innovations are mainly in the form of attacks rather than in the technical approach and theoretical aspects.

**Strength And Weaknesses:**

Strength:
1. the novelty and effectiveness of the proposed method.
2. the paper conducts a large number of defense experiments and analyzes the algorithm effect from different defense methods.
3. the method proposed in the paper can successfully achieve backdoor attacks with a lower poisoning ratio.
4. The paper is written in a clear and easy-to-understand manner.

Weaknesses:
1. although this paper uses formal data symbolic description for the proposed method, there is still no framework diagram to help the method understanding, which makes the algorithm of the article slightly inferior in the narration and implementation process.
2. although this paper introduces various attack methods in detail, it does not show more attack methods in experimental comparison, such as ISSBA. as a novel attack method, the authors should give more experimental comparison and analysis of the attack.
3, the author mentioned in the paper the advantages of the algorithm can also be mentioned in the attack on high efficiency. But for this part, I don't seem to see more theoretical analysis (convergence) and related experimental proofs. I have reservations about this point.
4. What is the main difference between the authors and ISSBA in terms of the formulation of the method? I would like the authors further to explain the contribution in conjunction with the formulas.

Some Questions:
1.How is the computational efficiency of extracting the trigger? Unlike previous backdoor attack algorithms, the method needs to analyze and extract data from the entire training dataset. Does this result in exponential time growth as the dataset increases?
2. The effectiveness and problem of the algorithm are that it requires access to the entire training dataset. Have the authors considered how the algorithm should operate effectively when the training dataset is not fully perceptible?

Overall:
The trigger proposed in this paper is novel, but the related validation experiments are not comprehensive, and the time complexity of the computation and the efficiency of the algorithm are not clearly analyzed. In addition, I expect the authors to further elucidate the technical contribution rather than the form of the attack.

**Summary Of The Paper:**

This paper proposes an algorithm for extracting backdoor triggers from clean training datasets to bypass existing backdoor defense methods due to the difficulty of defense posed by the fact that triggers are also widely present in clean samples. In addition, experiments show that this form of trigger can achieve a higher success rate with a better poisoning ratio.

**Summary Of The Review:**

From the current understanding, this article still has no solved efficiency problem and validation of effectiveness. Also, the innovativeness still needs further clarification. Of course, I am looking forward to the author's reply to me at the rebuttal stage.

---

### Official Review · Reviewer_nDYn · 2022-10-24

**Confidence:** 3
**Correctness:** 4
**Technical Novelty And Significance:** 4
**Empirical Novelty And Significance:** 3
**Recommendation:** 6

**Clarity, Quality, Novelty And Reproducibility:**

### Clarity
Overall, the paper is easy and clear to read. One issue is \sigma in Equation 4 was not defined.

### Quality
The paper is well written.

### Novelty
The idea is novel.

### Reproducibility
Code is provided.


**Strength And Weaknesses:**

### Strengths
- The proposed idea is novel and interesting. This proposal both improves the efficiency of the attack and increases the difficulty or cost of identifying and mitigating the trigger pattern.
- The experiments well proves the effectiveness of the proposed algorithm.
- The authors made a fair argument in the paragraph at the end of Page 6. They acknowledged that comparing backdoor methods by using their attack performance from a specific (default) configuration may be not so rigorous.

### Weaknesses
- In Equation 4, \sigma is not defined.
- The authors reported BA x 100, ASR x 100. It is preferable to show as percentage (%)
- What was the reason to set the target class as 2 instead of 0 or 1?
- I would love to see the results when using frequency-based backdoor defense [1]
[1]. Zeng Y, Park W, Mao ZM, Jia R. Rethinking the backdoor attacks' triggers: A frequency perspective. In ICCV 2021 (pp. 16473-16481).


**Summary Of The Paper:**

The paper proposes a novel backdoor attack that is efficient and stealthy. While the previous methods employ patterns that rarely occur
in benign data as the trigger pattern, this paper suggests using patterns that frequently appear in benign data of the target class but rarely appear in other classes. This proposal both improves the efficiency of the attack and increases the difficulty or cost of identifying and mitigating the trigger pattern. The proposed attack shows good attack performance on four popular datasets and passes common backdoor defenses.

**Summary Of The Review:**

Overall, the idea is novel and the paper is well written. I have some minor comments and I would love to see more defense results, particularly with the frequency-based defense.

---

> ### Author Response · Authors · 2022-11-08
> **Thank you for your careful review. Here are our answers.**
>
> Thank you for your kind response. We hope that our response would make sense to you. We would address your concerns below:
>
> 1. **Regarding the concern about the definition of $\sigma$ in Equation 4:**  \
> It is a pity that we forgot to give the definition of  $\sigma$ in the equation. It denotes the element-wise sigmoid activation function. We will update our manuscript to fix this mistake in the following.
>
> 2. **Regarding the concern about the setting of the target class:** \
> There is no special reason to set class 2 as the target class. Actually, we started our implementation based on a published source code. Class 2 is the default value of the target class in the code. We did not try other values of the target class since we think such a study is not essential to our claim.
>
> 3. **Regarding the concern about the performance of our method against frequency-based backdoor defense methods:** \
> To answer the question, we performed an experimental study based on the defense method proposed by [Zeng et. al, 2021] to evaluate the performance of our method against the frequency-based defense methods. As for the implementation, we follow the implementation of the published code (https://github.com/YiZeng623/frequency-backdoor/blob/main/Sec4_Frequency_Detection/Train_Detection.ipynb). We report the Accuracy (Acc) and Backdoored data Detection Rate (BDR) values of our method as well as BadNet for an intuitive comparison. As can be seen from the following table, our method achieves much smaller values of Acc and BDR against the frequency-based detector. This verifies the stealthiness of our method against the frequency-based defense methods. We will upload our implementation of the experiment in the following. We explain this phenomenon as follows: The frequency-based defense methods build on the assumption that the trigger pattern will result in severe high-frequency artefacts in the frequency domain. However, our method uses frequently occurring patterns in the target class as trigger patterns.  Such patterns should lie in the low-frequency region in the frequency domain, violating the assumption of the frequency-based defense methods. This is why our method can escape the defense of the frequency-based defense methods.
>
> ```
> | Dataset		| Acc		| BDR
> ---------------------------------------------------
> | MNIST (Our)		| 0.8754	| 0.7742
> | MNIST (BadNet)	| 0.9883	| 1.0
> ---------------------------------------------------
> | CIFAR10 (Our)		| 0.5588	| 0.1198
> | CIFAR10 (BadNet)	| 0.9989	| 1.0
> ---------------------------------------------------
> | GTSRB (Our)		| 0.6140	| 0.2354
> | GTSRB (BadNet)	| 0.9913	| 1.0
> ---------------------------------------------------
> | CelebA (Our)		| 0.5082	| 0.1899
> | CelebA (BadNet)	| 0.9986	| 1.0
> ```

---

### Decision · Program_Chairs · 2023-01-20

**Decision:**

Reject

**Justification For Why Not Higher Score:**

similarity to previous work, insufficient experiments.

**Justification For Why Not Lower Score:**

N/A

**Metareview: Summary, Strengths And Weaknesses:**

This work proposed to extract a pattern from benign target class data as the trigger, which is then embedded into the benign data through encoder to generate poisoned data.

There are several important concerns about this work, mainly including:
1. The highly similarity to a previous work ISSBA. The authors claimed the main difference is the trigger, one is extracted from benign data, while the other not. Some reviewers and I don't think it is a significant contribution, and the authors didn't provide convincing analysis about why such a trigger could improve the attack performance. Besides, the rebuttal claimed the proposed method could give higher ASR than ISSBA in the case of low poisoning rate. As one reviewer indicated, the possible reason may be the proposed trigger is more visible. Other comments about ISSBA in the rebuttal are also not well verified.
2. Inadequate experiments. Many recent attacks and defense methods are not compared. Although partial results are added in the rebuttal, they are insufficient to support the proposed method. For example, the results against frequency-based backdoor defense methods cannot tell the good performance against this defense.
3. There are also some suggestions about the writing, the analysis about the efficiency, as well as the threat model.

Overall, this work cannot be recommended as accept.